# Improved Survival Outcomes with Surgical Resection Compared to Ablative Therapy in Early-Stage HCC: A Large, Real-World, Propensity-Matched, Multi-Centre, Australian Cohort Study

**DOI:** 10.3390/cancers15245741

**Published:** 2023-12-07

**Authors:** Jonathan Abdelmalak, Simone I. Strasser, Natalie Ngu, Claude Dennis, Marie Sinclair, Avik Majumdar, Kate Collins, Katherine Bateman, Anouk Dev, Joshua H. Abasszade, Zina Valaydon, Daniel Saitta, Kathryn Gazelakis, Susan Byers, Jacinta Holmes, Alexander J. Thompson, Dhivya Pandiaraja, Steven Bollipo, Suresh Sharma, Merlyn Joseph, Amanda Nicoll, Nicholas Batt, Rohit Sawhney, Myo J. Tang, John Lubel, Stephen Riordan, Nicholas Hannah, James Haridy, Siddharth Sood, Eileen Lam, Elysia Greenhill, Ammar Majeed, William Kemp, John Zalcberg, Stuart K. Roberts

**Affiliations:** 1Department of Gastroenterology, Alfred Health, Melbourne, VIC 3004, Australia; j.abdelmalak@alfred.org.au (J.A.); tangmyojin@gmail.com (M.J.T.); j.lubel@alfred.org.au (J.L.); a.majeed@alfred.org.au (A.M.); w.kemp@alfred.org.au (W.K.); 2Department of Medicine, Central Clinical School, Monash University, Melbourne, VIC 3004, Australia; eileen.lam@monash.edu (E.L.);; 3Department of Gastroenterology, Royal Prince Alfred Hospital, Camperdown, Sydney, NSW 2050, Australia; simone.strasser@health.nsw.gov.au (S.I.S.); natalie.ngu@health.nsw.gov.au (N.N.); claude.dennis@health.nsw.gov.au (C.D.); 4Department of Gastroenterology, Austin Hospital, Heidelberg, VIC 3084, Australia; marie.sinclair@austin.org.au (M.S.); avik.majumdar@austin.org.au (A.M.); kate.collins3@austin.org.au (K.C.); kat.bateman@austin.org.au (K.B.); 5Department of Gastroenterology, Monash Health, Clayton, VIC 3168, Australia; anouk.dev@monash.edu (A.D.); joshua.abasszade@monashhealth.org (J.H.A.); 6Department of Gastroenterology, Western Health, Footscray, VIC 3011, Australia; zina.valaydon@wh.org.au (Z.V.); daniel.saitta@wh.org.au (D.S.); kathryn.gazelakis@wh.org.au (K.G.); susan.byers1@wh.org.au (S.B.); 7Department of Gastroenterology, St. Vincent’s Hospital Melbourne, Fitzroy, VIC 3065, Australia; jacinta.holmes@svha.org.au (J.H.); alexander.thompson@svha.org.au (A.J.T.); dhivya.pandiaraja@svha.org.au (D.P.); 8Department of Medicine, St. Vincent’s Hospital, University of Melbourne, Parkville, VIC 3052, Australia; 9Department of Gastroenterology, John Hunter Hospital, New Lambton Heights, NSW 2305, Australia; steven.bollipo@health.nsw.gov.au (S.B.); suresh.sharma@health.nsw.gov.au (S.S.); merlyn.joseph@health.nsw.gov.au (M.J.); 10Department of Gastroenterology, Eastern Health, Box Hill, VIC 3128, Australia; amanda.nicoll@easternhealth.org.au (A.N.); nicholas.batt@easternhealth.org.au (N.B.); rohit.sawhney@easternhealth.org.au (R.S.); 11Department of Medicine, Eastern Health Clinical School, Box Hill, VIC 3128, Australia; 12Department of Gastroenterology, Prince of Wales Hospital, Randwick, NSW 2031, Australia; stephen.riordan@health.nsw.gov.au; 13Department of Gastroenterology, Royal Melbourne Hospital, Parkville, VIC 3052, Australia; nicholas.hannah2@mh.org.au (N.H.); james.haridy@mh.org.au (J.H.); sood.s@unimelb.edu.au (S.S.); 14School of Public Health and Preventive Medicine, Monash University, Melbourne, VIC 3004, Australia; john.zalcberg@monash.edu; 15Department of Medical Oncology, Alfred Health, Melbourne, VIC 3004, Australia

**Keywords:** hepatocellular carcinoma, early, resection, ablation

## Abstract

**Simple Summary:**

Cure is the goal of treatment in early primary liver cancer with surgical resection and ablation therapy being the two most common modalities used. This real-world multi-centre Australian study demonstrates that surgical treatment results in superior outcomes. We observed a significantly reduced risk of death from any cause and of recurrent liver cancer after controlling for factors such as initial tumour burden, liver disease severity and other medical comorbidities. Our study provides compelling evidence to recommend surgery for suitable patients to achieve the best possible outcomes.

**Abstract:**

The optimal treatment approach in very-early and early-stage hepatocellular carcinoma (HCC) is not precisely defined, and there is ambiguity in the literature around the comparative efficacy of surgical resection versus ablation as curative therapies for limited disease. We performed this real-world propensity-matched, multi-centre cohort study to assess for differences in survival outcomes between those undergoing resection and those receiving ablation. Patients with Barcelona Clinic Liver Cancer (BCLC) 0/A HCC first diagnosed between 1 January 2016 and 31 December 2020 who received ablation or resection as initial treatment were included in the study. A total of 450 patients were included in the study from 10 major liver centres including two transplant centres. Following propensity score matching using key covariates, 156 patients were available for analysis with 78 in each group. Patients who underwent resection had significantly improved overall survival (log-rank test *p* = 0.023) and local recurrence-free survival (log rank test *p* = 0.027) compared to those who received ablation. Based on real-world data, our study supports the use of surgical resection in preference to ablation as first-line curative therapy in appropriately selected BCLC 0/A HCC patients.

## 1. Introduction

Hepatocellular carcinoma (HCC) worldwide is both common and deadly, accounting for 830,000 deaths in 2020 [1] and with an incidence that is expected to continue rising over the coming years [2]. HCC screening is performed in at-risk patients to detect HCC in its early stages when curative treatment can still be offered. The Barcelona Clinic Liver Cancer (BCLC) staging system is the most commonly used management algorithm, with BCLC 0 or very-early disease referring to a single tumour, 2 cm or less, associated with preserved liver function (Child–Pugh (CP)A) and cancer-related performance status of Eastern Cooperative Oncology Group (ECOG) 0. BCLC A or early-stage HCC describes patients with a single tumour of any size, or up to 3 tumours with the largest 3 cm or less, with relatively preserved liver function (CPA or B) and ECOG 0.

For BCLC 0 disease, the most recently updated BCLC treatment strategy [3] recommends ablation for non-transplant candidates, resection for transplant candidates without clinically significant portal hypertension (CSPH) and normal bilirubin, and upfront transplantation for patients with CSPH or increased bilirubin [3]. For those with BCLC A disease with a single tumour, resection is recommended for those with good liver function in the absence of CSPH, while those with CSPH and/or elevated bilirubin should proceed to upfront transplantation or ablation if contraindications to transplantation are present [3]. Similarly, in BCLC A disease with two or three nodules, patients are recommended for upfront transplantation with ablation as an alternative if the patient is not a transplant candidate [3].

In clinical practice, however, access to liver transplantation, which is ultimately a cure for both HCC and the underlying liver disease, is significantly limited by organ availability, cost and long-term health implications. Australian [4] and other national guidelines [5,6,7] have a lesser emphasis on transplantation and generally recommend resection as first-line therapy for all patients with new diagnosis of BCLC 0/A disease, including those with more than one lesion, providing CSPH is absent and there is predicted sufficient liver remnant post-surgery in the context of the underlying liver disease. Ablation is recommended as an alternative treatment modality for patients with BCLC 0/A disease when resection is not feasible and transplant is not imminently considered.

Nevertheless, over the last two decades, percutaneous ablation with thermal techniques such as radiofrequency ablation (RFA) and microwave ablation (MWA) has emerged as a suitable alternative treatment modality to surgical resection in those with limited disease, particularly those with borderline liver function. Indeed, multiple studies [8,9,10,11,12] have failed to show a significant difference in overall survival between resection and ablation, and results of published meta-analyses comparing the two treatments in BCLC 0, 0/A and A disease have similarly had mixed results, potentially due to poor trial design in many of the primary studies [13,14,15,16,17,18,19,20,21,22,23,24].

In the context of this ongoing debate, we performed this study to assess if, in a real-world Australian cohort of BCLC 0/A HCC patients, there is a significant difference in survival outcomes between surgical resection and ablative therapy in order to better inform treatment decisions in this at-risk patient population.

## 2. Materials and Methods

### 2.1. Participants

This study involved participants with a diagnosis of HCC between 1 January 2016 and 31 December 2020 at two Australian Liver Transplant and HCC quaternary centres and a further eight Australian HCC tertiary referral liver centres across Victoria and New South Wales. Patients were eligible for the study if they met the following inclusion criteria: adult aged > 18 years of age; diagnosis of HCC documented between 1 January 2016 and 31 December 2020 on the basis of imaging fulfilling LIRADS-5 criteria or histology confirming HCC; confirmed BCLC 0 or A disease based on single lesion of any size or up to 3 lesions with no lesions > 3 cm, CP A or B, cancer-related performance status of ECOG 0, absence of extrahepatic disease or vascular invasion; and received curative-intent therapy with either surgical resection or ablative therapy including microwave ablation (MWA) or radiofrequency ablation (RFA). Exclusion criteria were prior diagnosis or past history of HCC; diagnosis of other solid organ malignancy other than non-melanotic skin cancer and insufficient data in the medical record to adequately describe stage of HCC.

Waiver of consent was sought with all patient data entered in a deidentified form. Ethics for the study was approved by the Monash Health Human Research Ethics Committee (HREC).

### 2.2. Study Design

This was a multi-centre retrospective cohort study. Data were collected retrospectively from the medical record, from the date of initial diagnosis of HCC to the end of follow-up (either death or last medical record entry available at time of data extraction). Data regarding demographic, clinical, biochemical and tumour characteristics were collected along with relevant treatment and outcome data. Modified RECIST criteria (mRECIST) [25] were used at all sites to describe treatment response post initial treatment and at subsequent follow-up with ‘Complete Response’ (CR) defined as the disappearance of arterial enhancement within all target lesions. The minimum dataset is outlined in Appendix B. Data were deidentified and entered into a centralised REDCap electronic data capture tool hosted at Monash University.

### 2.3. Endpoints

The primary endpoint was local recurrence-free survival (LRFS) which is defined as the time from documented cure to either death or documented local recurrence. The date of resection or the date of the first documented complete response after ablative therapy was considered the index date. Secondary endpoints of interest were (a) recurrence-free survival (RFS), (b) overall survival (OS), which is defined as time from diagnosis to death, and (c) liver-related survival (LRS), which is defined as time from diagnosis to liver-related death (with non-liver death considered a censoring event). Rates of attainment of CR for patients who received ablation were also reported and notably, patients who failed to ever achieve CR were excluded from LRFS and RFS analysis.

We used LRFS/RFS rather than disease-free survival (DFS) to prevent failed attempts at ablation to significantly skew the results. LRFS was chosen as the best indicator of local tumour control, which is the goal of curative treatment in early-stage HCC, as late non-local recurrence is likely driven by de novo hepatocarcinogenesis rather than a failure of curative therapy. Due to the appropriate role for transplant as a follow-up curative treatment for recurrent HCC, concern that transplant would otherwise significantly skew the results, and our desire to assess real-world impact of the initial treatment decision irrespective of future follow-up treatment, OS and LRS was analysed without transplantation considered a censoring event. Lastly, major complications, defined as a treatment-related adverse event resulting in escalation in medical care, prolonged hospitalisation or death, were also reported.

### 2.4. Statistical Analysis

Data were analysed by SPSS 29.0 software (SPSS, Inc., Chicago, IL, USA). Binary logistic regression, using forward-selection strategy, was used to determine the factors predicting treatment group allocation (resection or ablation). Results of regression analysis are presented in Appendix A. The following variables were used to calculate the propensity score: age, sex, management at transplant-centre versus non-transplant centre, diabetes, smoking, HBV and alcohol as cause of background liver disease, tumour burden category (single tumour ≤ 2 cm, single tumour 2 to ≤3 cm, single tumour > 3 cm or multiple tumour with largest < 3 cm), platelet count, CP score and Charlson Comorbidity Index (CCI). Nearest-neighbour propensity score matching in a 1:1 ratio, with a match tolerance set at 0.01, was then performed. Match tolerance was initially set at 0.1 and was systematically reduced to find the highest value where all variables of interest were adequately matched between groups.

The statistical significance of differences between the two groups before and after propensity score matching was performed using a Chi-square test for categorical characteristics, Mann–Whitney U-test for non-parametric variables and independent sample *t*-test for parametric variables. Similarly, a histogram of propensity scores was constructed to ensure that matching had been successful.

In the matched cohort, Kaplan–Meier analysis was used to assess LRFS, RFS, OS, and LRS in the ablation and resection groups with a log-rank test used to ultimately assess for statistically significant differences between the two groups. Point survival rates at 1- and 3-year follow-up were also calculated with a log-rank test used to assess for the significance of survival differences up until these specified timepoints.

In the event of finding a significant difference in LRFS, RFS, OS or LRS, further Kaplan-Meier analysis was performed in the original unmatched cohort to ensure that the findings were reproducible outside of the propensity-score matched conditions.

In all tests of statistical significance performed, a two-tailed *p* < 0.05 was deemed as a statistically significant difference.

## 3. Results

### 3.1. Patients

A total of 450 patients met eligibility criteria and were included in the study with 254 in the ablation group (RFA = 49, MWA = 205) and 196 in the resection group. Figure 1 summarises the study design.

Prior to matching, resection patients were systematically different to those who underwent ablation. In particular, treatment allocation to resection was associated with a number of more favourable prognostic indicators including younger age, higher rates of hepatitis B and lower rates of alcohol, fewer medical comorbidities with lower CCI, lower rates of diabetes and lastly, significantly greater platelet counts and lower CP score, indicating a lesser likelihood of cirrhosis and portal hypertension. In contrast, resection patients were likely to have larger single tumours in comparison to a preponderance of small single tumours in the ablation group. Because of this, resection patients were more likely to have their disease classified as stage BCLC A in comparison to the higher rates of BCLC 0 disease in those who underwent ablation.

After propensity score matching, 78 matched pairs for a total of 156 patients were produced. In the matched cohort, there were no significant differences seen between the two groups. Patient characteristics before and after matching are outlined in detail in Table 1. Notably, in the matched cohort, a total of 74 out of 156 patients had BCLC 0 disease (CPA and single lesion 2 cm or less) and 149 out of 156 patients had CPA disease. Figure 2 shows the similar distribution of propensity scores in the two matched groups in comparison to the significant differences prior to matching in the original cohort.

### 3.2. Outcomes

Outcomes in the original unmatched cohort are presented in Table 2 alongside the outcomes seen in the PSM cohort. In the original unmatched cohort, there was a significant number of patients (32 out of 254, 12.6%) who failed to achieve CR with ablation as the initial treatment strategy. A further 31 patients (12.2%) required more than one ablation to achieve CR. There were no major complications seen in the ablation group in contrast to two cases in the resection group, representing a 1.0% major complication rate. Only a small number of patients underwent transplantation during their follow up. In all these patients, this occurred after HCC recurrence.

### 3.3. Recurrence-Free Survival

In the matched cohort, over the entire period of recorded follow-up (median follow-up time 37.9 months or 1136 days), there was a non-significant trend towards improved RFS in the resection group compared to ablation (log rank test *p* = 0.068). Survival curves (shown in Figure 3) show a clear separation in RFS between 3 and 36 months; however, there is a subsequent high number (*n* = 9) of non-local recurrences in the resection group, bringing the two curves closer together. There was indeed a clear 3-year recurrence-free survival benefit seen with resection with RFS rates of 75.6% vs. 57.5% (*p* = 0.007). One-year recurrence-free survival was 92.3% in the resection group versus 83.6% (*p* = 0.091).

Unadjusted analysis performed in the original unmatched cohort showed similar significant difference in recurrence-free survival with superiority seen in the resection group (log rank *p* < 0.001) and similar 1- and 3-year recurrence-free survival rates (88.3% vs. 77.5%, *p* = 0.003; 73.0% vs. 50.9%, *p* < 0.001, respectively). Kaplan–Meier survival curves representing the entire original unmatched cohort are shown in Appendix A.

### 3.4. Local Recurrence-Free Survival

In the original unmatched cohort, 45 out of 105 recurrences (42.9%) in the ablation group occurred locally at the site of the ablation zone compared to 14 out of 63 recurrences (22.2%) in the resection group (*p* = 0.007). Similarly, in the matched cohort, there was a higher proportion of ablation patients with recurrent tumours at the site of previous treatment compared to resection patients (13 out of 33 (39.4%) compared to 7 out of 30 (23.3%), although this was not statistically significant *p* = 0.171).

Accordingly, the difference in local recurrence-free survival between the two propensity-matched groups is more pronounced than the difference in recurrence-free survival with Kaplan–Meier LRFS survival curves presented in Figure 4 showing a significant difference between the two groups (log rank test *p* = 0.027) with most of the separation of the two curves occurring between 3 and 24 months, representing the high number of local recurrences occurring in the ablation group over this time. One-year local recurrence-free survival was 97.4% vs. 90.3% (*p* = 0.067). There was a significant difference seen in 3-year LRFS rates (91.0% vs. 79.5%, *p* = 0.028).

Sensitivity analysis was performed in the matched cohort with the exclusion of those with a single tumour >3 cm, and the superiority of the resection group was again demonstrated (log rank test 0.028, Appendix A). The significant difference in LRFS was also observed in the original unmatched cohort (Appendix A, log rank test *p* < 0.001).

### 3.5. Overall Survival

Overall survival was superior in the resection group compared to the ablation group in the matched cohort (log rank test *p* = 0.023) with survival curves presented in Figure 5 demonstrating a separation in curves occurring mainly from 24 to 48 months where most deaths were recorded. The median overall follow-up time was 53.3 months (1598.5 days). While the overall survival difference over the entire follow-up was significantly different (*p* = 0.023), the difference was not significant at the 1-year and 3-year timepoints (100% vs. 97.4%, *p* = 0.159; 97.4% vs. 91.0%, *p* = 0.071, respectively) with higher numbers of deaths in the ablation group.

Sensitivity analysis performed in the matched cohort with the exclusion of patients with tumours >3 cm demonstrated a non-significant trend towards improved survival (log rank test *p* = 0.100, Appendix A). Unadjusted survival rates in the original unmatched cohort across resection and ablation groups were similar to those in the matched cohort and statistically significant in the larger cohort (1-year survival 99.0% vs. 96.0%, *p* = 0.058; 3-year survival 95.4% vs. 87.0%, *p* = 0.002) with an overall survival benefit also seen in comparing the two Kaplan–Meier survival curves over the entirety of follow-up (Appendix A, log rank test *p* < 0.001).

### 3.6. Liver-Related Survival

In the propensity-matched cohort, 7 out of 11 deaths (63.6%) in the ablation group were liver-related compared to 2 out of 3 (66.7%) in the resection group. Similar proportions were seen in the original unmatched cohort (29 out of 41 liver-related deaths (70.7%) in ablation group, 7 out of 10 liver-related deaths (70.0%) in resection group). On performing Kaplan–Meier survival analysis, there was a trend towards improved LRS in the resection group, which failed to reach statistical significance (log rank test *p*= 0.074) with most events occurring between 24 and 48 months (Figure 6).

## 4. Discussion

In clinical practice, the management approach to BCLC 0/A HCC is complex, nuanced, and individualised. Several factors including the severity and aetiology of liver disease, age, non-liver comorbidities, potential candidacy for liver transplantation, tumour number, size and location together with the views and values of the patient are all key elements involved in the decision-making process by the multi-disciplinary teams managing these patients. Decision making is further complicated by conflicting evidence. While international and national guidelines generally recommend hepatic resection in preference to ablation in BCLC 0/A HCC, there is mixed evidence in the literature regarding if and to what extent the outcomes differ between the two curative treatment modalities. 

A recently published meta-analysis suggested resection achieves superior OS and PFS in all patients with BCLC A disease with multiple lesions or single lesion > 3 cm, but they found no difference in BCLC 0 patients and those with BCLC A disease with a single tumour ≤ 3 cm [26], thus calling into question the superiority of resection in those with a single small HCC. In contrast, two previously published meta-analyses [22,23] showed resection was associated with superior RFS and OS in patients exclusively with BCLC 0 disease with solitary tumours ≤ 2 cm. Earlier meta-analyses [13,14,16,17,18] similarly show contradictory results, highlighting the uncertainty around whether or not resection is truly associated with superior outcomes compared to ablation. Contraindications to resection in early-stage disease have also been challenged with a recent study demonstrating that localised hepatic resection for HCC is safe in those with mild CSPH or mildly elevated bilirubin in appropriately selected patients [27].

We therefore performed this study in order to examine if, in a real-world Australian cohort of BCLC 0/A HCC patients, there was a discernible difference in real hard endpoints, such as recurrence and death, between those receiving surgical resection and ablative therapy. In a propensity-matched cohort comprising 156 patients, we found that there was indeed a significant improvement in local recurrence-free survival and overall survival associated with hepatic resection compared to ablation. Our findings provide a valuable addition to the existing literature, demonstrating the superiority of resection in a real-world combined cohort of BCLC 0/A patients, providing an evidence base of support for the recommendation of resection where possible in patients with early/very early-stage disease.

As expected, we found systematic differences in the original two groups prior to matching particularly with respect to liver disease severity and tumour burden. Notably, almost all patients undergoing resection had compensated Child–Pugh A5 or 6 liver disease and preserved platelet count compared to ablation patients who were significantly more likely to have Child–Pugh B liver disease and thrombocytopenia as a marker of CSPH. Patients in the resection group were more likely to have a larger solitary tumour than those in the ablation group (78 out of 196 vs. 15 out of 254). This is not unexpected, as we only included BCLC 0/A patients who underwent either resection or ablation alone as first-line therapy rather than those who received combination therapy with TACE followed by ablation. While there was a preponderance of small single tumours in the ablation group (122 out of 254 compared to 50 out of 196), patients undergoing ablation were also more likely to have more than one HCC (42/254 vs. 10/196). The third major difference was in patient demographics with patients undergoing resection on average being younger and less comorbid, as indicated by CCI and age. In patients with BCLC 0 disease in particular, advanced age and non-liver comorbidities are often a compelling reason to pursue ablation rather than resection, and this likely explains the systematic difference. Lastly, patients undergoing resection were more likely than ablation patients to be receiving treatment at a transplant centre. It is therefore possible that transplant centres were systematically more likely to offer resection to borderline candidates for surgery compared to non-transplant centres, which was perhaps due to the experience of the specialist liver transplant surgeons at these centres.

To accurately assess for the impact of the treatment alone in affecting outcomes, we utilised propensity score matching to attempt to minimise the effect of these confounding factors, particularly given the significant systematic differences between the two groups. Propensity score matching is a quasi-experimental technique that aims to minimise the effects of confounding in observational studies by making each of the two treatment groups as similar as possible based on the other extraneous variables. Utilising propensity score matching, we produced a cohort of 78 matched pairs (total 156 patients) with all systematic differences eliminated post-matching. We then performed Kaplan–Meier survival analysis with a log-rank test in the propensity matched cohort assessing for differences in recurrence-free survival, local recurrence-free survival, overall survival and liver-related survival. We found that patients undergoing resection had significantly improved local recurrence-free survival (log rank test *p* = 0.027) with this translating to improved 3-year recurrence-free survival (log rank test *p* = 0.007), suggesting superior local tumour control with resection. Interestingly, we found that beyond four years of follow up, the resection group had a significant number of non-local HCC recurrence. These events are likely to represent true de novo tumours, although the possibility of slow-growing intrahepatic metastasis presenting late cannot be excluded.

Importantly, we found that patients undergoing resection had significantly superior overall survival (log rank test *p* = 0.023) with reduced mortality and separation of curves noted at 24–48 months since diagnosis. This is a remarkable finding, and it highlights that in real-world practice, even after controlling for liver disease severity and non-liver comorbidities, resection offers a survival advantage compared to ablation. Of note, however, our sensitivity analysis in the matched cohort with tumours ≤ 3 cm failed to show a significant difference (*p* = 0.100), but this is not surprising given the overall small number of event numbers in this cohort (seven deaths in the ablation group, two in the resection group).

Many of the deaths occurring in the ablation group occurred in those who failed to achieve CR with ablation as well as those who developed recurrent disease, highlighting the importance of achieving disease control to maximise survival. Most deaths occurred at 24–48 months from diagnosis and were liver-related, highlighting the need for timely and effective strategies at diagnosis to reduce mortality risk. It is unlikely that transplantation, as a competing event, has confounded our results, with our cohort matched on age, CCI and liver disease severity—common factors determining suitability for transplant referral—and overall, only a small number of transplants were performed during follow-up with slightly higher rates in the ablation group (4/78 vs. 2/78).

Despite the preponderance of liver-related death (two out of two in the resection group, seven out of nine in the ablation group), we failed to show a significant difference between the two groups (*p* = 0.074), which was likely due to the overall small event numbers. With greater patient numbers, or longer follow-up for the subset of patients censored before 36–48 months, we might expect to find a significant difference. We chose not to consider transplant as a competing event with liver-related death, as we cannot say with certainty that patients who underwent transplant as treatment for recurrent HCC would have necessarily died during follow-up without transplantation. Such patients may have reasonable survival outcomes with further locoregional treatment instead, as demonstrated in much of the cohort with recurrent HCC who did not undergo transplantation.

A major concern balancing against the utility of resection as a curative treatment for early-stage HCC in comparison to ablation is the risk of surgical complications. However, encouragingly, in our study, the major complication rate associated with resection was seen to be low (1.0%) in contrast to previously published data [16,24], which was potentially due to the real-world nature of our study in which patients were carefully selected for resection in the context of multidisciplinary discussion. With the observed positive impact on overall survival and recurrence rates, our study provides compelling support for resection where possible in preference to ablation for patients with BCLC 0/A HCC.

Our study has several strengths. Firstly, it involved real-world data where individualised patient decisions were made by 10 distinct multidisciplinary teams across Australia, allowing for an assessment of the impact of treatment allocation to resection versus ablation that is framed within the real complexities and nuance of everyday clinical practice. Secondly, we used propensity score matching as part of the study design, which increases confidence that the observed difference in outcomes between the two groups is due to the treatments themselves rather than any systematic difference in confounders between the two groups. Lastly, we had a sufficiently large patient cohort such that even after propensity score matching and loss of unmatched cases, there was appropriate statistical power to observe a statistically significant difference in overall survival and local recurrence-free survival between the two groups.

Our study does, however, have significant major limitations. Firstly, our data are retrospectively collected and solely observational, and this increases the risk of selection bias, information bias and confounding. However, this is partially mitigated against with the use of propensity score matching in artificially eliminating systematic differences in covariates (including those predicting unsuitability for resection such as severe thrombocytopenia associated with portal hypertension, advanced age and significant medical comorbidities) between groups as well as the use of sensitivity analysis in the original unmatched population to ensure all patient data have been assessed. Due to the limitations of the data capture, nuanced assessment of tumour location and the implications on resectability were not able to be assessed, which may introduce a component of unaddressed selection bias. Secondly, our study was limited in follow-up time. Median follow-up time for RFS and OS was 37.9 months and 53.3 months, respectively. We suspect that for the patients censored before 36 months, a sizeable proportion would go on to develop an event of interest, such as recurrence. Despite the limited follow-up time, we were still able to show a significant difference in LRFS and OS, and it is likely that the observed difference in outcomes would translate to longer periods of follow-up. It is, however, possible that differences in LRS would have become more pronounced with a larger number of events with either longer follow-up or greater patient numbers.

## 5. Conclusions

In a real-world cohort of Australian early-stage HCC patients, resection compared to ablation confers a significant overall survival benefit, which is likely driven by the superiority of resection in the durable achievement of local tumour control. Resection has a low risk of major complications in appropriately selected patients. Our study provides valuable evidence that resection should be offered in preference to ablation in suitable early-stage HCC patients.

## Figures and Tables

**Figure 1 cancers-15-05741-f001:**
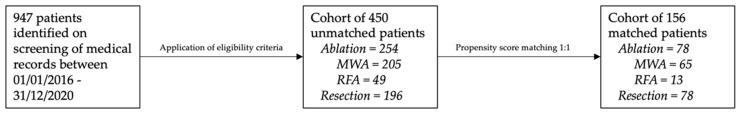
Summary of study design with number of patients before and after propensity score matching.

**Figure 2 cancers-15-05741-f002:**
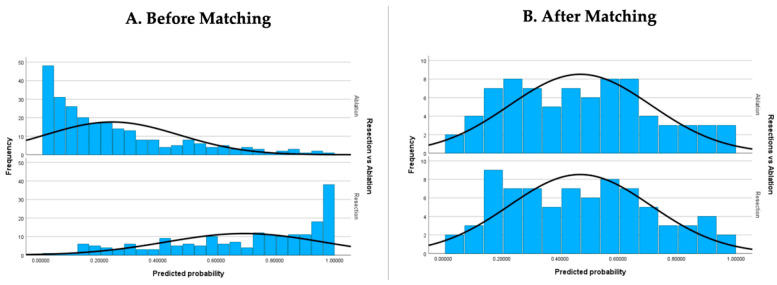
Distribution of propensity scores in resection and ablation groups before and after matching.

**Figure 3 cancers-15-05741-f003:**
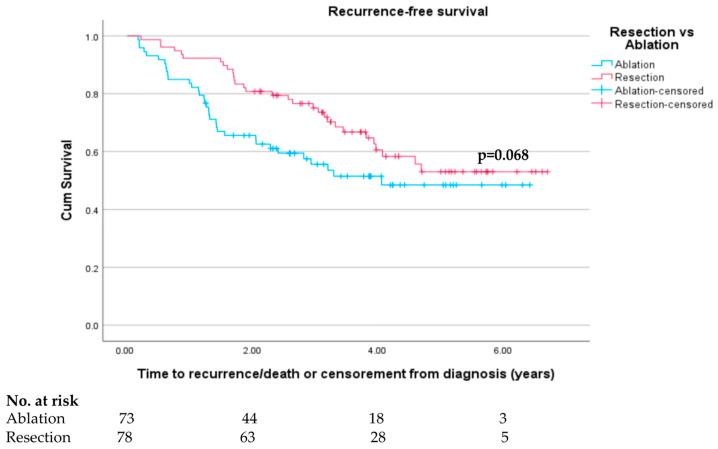
Recurrence-free survival in propensity-score-matched resection and ablation groups.

**Figure 4 cancers-15-05741-f004:**
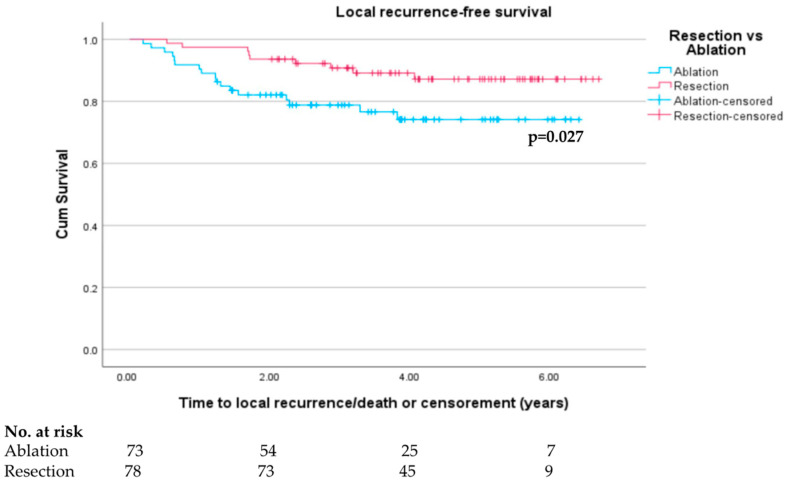
Local recurrence-free survival in propensity-score-matched resection and ablation groups.

**Figure 5 cancers-15-05741-f005:**
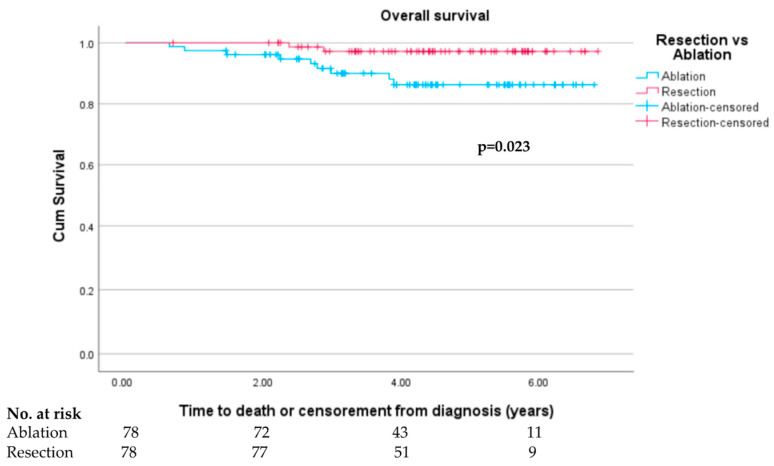
Overall survival in propensity-score matched resection and ablation groups.

**Figure 6 cancers-15-05741-f006:**
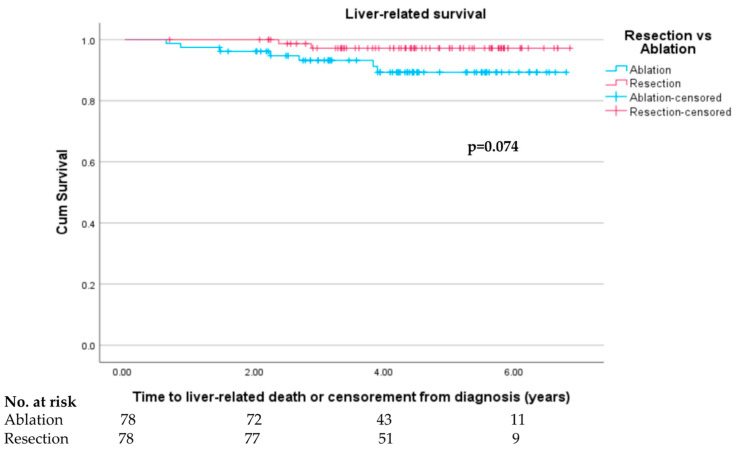
Liver-related survival in propensity-score-matched resection and ablation groups.

**Table 1 cancers-15-05741-t001:** Patient characteristics in ablation and resection groups before and after propensity score matching.

	Matched (*n* = 156)			Unmatched (*n* = 450)		
Characteristic	Ablation *n* = 78	Resection *n* = 78	*p*-Value	Ablation *n* = 254	Resection *n* = 196	*p*-Value
Gender Male Female	63 15	60 18	0.556	201 53	154 42	0.885
Age *	63.9 ± 8.1	63.0 ± 8.7	0.491	65.4 ± 9.9	63.3 ± 9.6	0.026
Transplant Centre No Yes	52 26	47 31	0.406	182 72	117 79	0.008
Aetiology Alcohol HBV HCV MASLD Other metALD HBV + HCV HCV + SLD HBV + SLD	7 10 21 5 2 0 2 25 6	8 11 19 5 2 4 4 21 4	0.689	49 24 41 34 9 12 10 67 8	18 50 35 23 13 6 5 36 10	<0.001
Diabetes Absent Present	64 14	63 15	0.837	179 75	165 31	<0.001
Smoking Absent Present	48 30	47 31	0.870	169 85	138 58	0.382
Platelet count **	156.5 (116–206)	142 (112–178)	0.280	116 (81–155)	182 (136.5–236.5)	<0.001
CCI **	3 (2–5)	4 (2–5)	0.390	5 (3–6)	3 (2–4)	<0.001
Tumour category Single lesion <2 cm 2–3 cm >3 cm >1 lesion	43 21 10 4	33 25 14 6	0.435	122 75 15 42	50 57 78 10	<0.001
Child–Pugh Score A5 A6 B7 B8 B9	58 17 2 1 0	56 18 3 1 0	0.967	124 83 31 10 6	164 26 4 2 0	<0.001
BCLC 0 A	42 36	32 46	0.109	101 153	49 147	<0.001
Ablation Modality RFA MWA	13 65			49 205		

HBV, Hepatitis B virus; HCV, Hepatitis C virus; MASLD, metabolic-dysfunction associated steatotic liver disease; metALD, metabolic and alcohol-related liver disease; CCI, Charlson Comorbidity Index; BCLC, Barcelona Clinic Liver Cancer; RFA, radiofrequency ablation; MWA, microwave ablation. * Mean ± standard deviation. ** Median (25th percentile–75th percentile).

**Table 2 cancers-15-05741-t002:** Summary of overall, 1- and 3-year outcomes in the PSM and original unmatched cohort.

	Matched Cohort			Unmatched Cohort		
Outcomes	Ablation *n* = 78	Resection *n* = 78	*p*-Value	Ablation *n* = 254	Resection *n* = 196	*p*-Value
CR Yes First ablation Subsequent ablation Never	73 (94.6%) 61 (76.2%) 12 (15.4%) 5 (5.4%)			222 (87.4%) 191 (75.2%) 31 (12.2%) 32 (12.6%)		
Events None Failure to achieve CR Recurrence Local Distant Death Liver-related Non-liver-related Transplant Major complication	39 (50.0%) 5 (5.4%) 33 (42.3%) 13 (16.7%) 20 (25.6%) 9 (11.5%) 7 (9.0%) 2 (2.6%) 4 (5.2%) 0	48 (61.5%) 30 (38.5%) 7 (9.0%) 23 (29.5%) 2 (2.6%) 2 (2.6%) 0 2 (2.6%) 0		106 (41.7%) 32 (12.6%) 105 (41.3%) 45 (17.7%) 60 (23.6%) 41 (16.1%) 29 (11.4%) 12 (4.7%) 9 (3.5%) 0	130 (66.3%) 63 (32.1%) 14 (7.1%) 49 (25.0%) 10 (5.1%) 7 (3.6%) 3 (1.5%) 4 (2.0%) 2 (1.0%)	
Recurrence-free survival At 1-year At 3-year follow-up At end of follow-up	83.6% 57.5% 53.4%	92.3% 75.6% 61.5%	0.091 **0.007** 0.068	77.5% 50.9% 47.7%	88.3% 73.0% 66.3%	**0.003** **<0.001** **<0.001**
Local recurrence-free survival At 1-year follow-up At 3-year follow-up At end of follow-up	90.3% 79.5% 76.7%	97.4% 91.0% 88.5%	0.067 **0.028** **0.027**	87.7% 73.0% 71.2%	97.4% 90.8% 89.3%	**<0.001** **<0.001** **<0.001**
Overall survival At 1-year follow-up At 3-year follow-up At end of follow-up	97.4% 91.0% 88.5%	100% 97.4% 97.4%	0.159 0.071 **0.023**	96.0% 87.0% 83.9%	99.0% 95.4% 94.9%	0.058 **0.002** **<0.001**
Liver-related survival At 1-year follow-up At 3-year follow-up At end of follow-up	97.4% 93.6% 91.0%	100% 97.4% 97.4%	0.159 0.217 0.074	97.2% 90.9% 88.6%	99.0% 96.4% 96.4%	0.192 **0.015** **0.001**

## Data Availability

The data presented in this study are available on request from the corresponding author. The data are not publicly available due to privacy concerns.

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
