# Peer review of "Improved Survival Outcomes with Surgical Resection Compared to Ablative Therapy in Early-Stage HCC: A Large, Real-World, Propensity-Matched, Multi-Centre, Australian Cohort Study"

_cancers, 2023, doi:10.3390/cancers15245741_

Round 1

Reviewer 1 Report

Comments and Suggestions for Authors

In this manuscript, “Improved Survival Outcomes with Surgical Resection Compared to Ablative Therapy in Early-Stage HCC: A Large, Real World, Propensity-Matched, Multi-Centre, Australian Cohort Study,” the authors assessed differences in survival outcomes between HCC patients receiving resection versus those receiving ablation as a treatment for HCC, in the early stages of the disease. They concluded that surgical treatment resulted in superior outcomes (patient survival and recurrence of liver cancer).

Major Comments

The superiority of resection vs. ablation as modalities for treating HCC remains debated. The choice of modality depends on the ability of the patient to tolerate surgery due to the size of the lesion, the liver residual function, and the multifocal distribution of tumor nodules. Although this study is in a large propensity-matched cohort, it is retrospective. Future randomized control trials are needed to determine the preferable modality for treating HCC.

Minor Comments

1.       Adding a diagram for the study design will be helpful to the reader.

2.       A flow chart of the findings before and after matching would strengthen the paper

Comments on the Quality of English Language

Some typos in the text

Author Response

  1. Adding a diagram for the study design will be helpful to the reader.

We have added Figure 1 – a flowchart which clearly illustrates the study design.

  1. A flow chart of the findings before and after matching would strengthen the paper

Currently patient characteristics before and after matching are summarised in Table 1 and outcomes before and after matching are summarised in Table 2. Because of the volume of data to present, we believe that a flow chart with this information would be difficult to read and that tables are the most appropriate format to present these findings

Reviewer 2 Report

Comments and Suggestions for Authors

THe study was well written and clearly presented. However, i have serious concerns on the novelty of the topic as several previous papers and even meta-analyses already covered the topic and led to similar results. Therefore, i don't see any points in conducting another study with the same findings.

Author Response

There are no comments to respond to

Reviewer 3 Report

Comments and Suggestions for Authors

This is a multicenter Austrarian retrospective study comparing ablation and resection for patients with early stage hepatocellular carcinoma. I have several comments.

1. What is "CR"? It should be spelled out at initial appearance.

2. In the results section, what is "systematically different"?

3. In the results section, no significant difference was seen for what point?  

Author Response

  1. What is "CR"? It should be spelled out at initial appearance.

We apologise for this omission – we have included an explanation in lines 137-138 and have also cited the mRECIST criteria earlier in the same sentence (Reference 25. Lencioni R, Llovet JM. Modified RECIST (mRECIST) assessment for hepatocellular carcinoma. Semin Liver Dis. 2010;30(1):52-60.)

  1. In the results section, what is "systematically different"? AND 3. In the results section, no significant difference was seen for what point?

We have now bolded the statistically significant p values in Table 2 to make the distinction between the significant and non-significant results clearer to the reader. In addition, we have made it clearer in the text in Para 1 of the results section what was systematically different between the two treatment groups prior to matching.